# Differential Glial Response and Neurodegenerative Patterns in CA1, CA3, and DG Hippocampal Regions of 5XFAD Mice

**DOI:** 10.3390/ijms252212156

**Published:** 2024-11-12

**Authors:** Tahsin Nairuz, Jin-Chul Heo, Jong-Ha Lee

**Affiliations:** Department of Biomedical Engineering, Keimyung University, Daegu 42601, Republic of Korea; tahsin.bmb@nstu.edu.bd (T.N.); washingbuffer@gmail.com (J.-C.H.)

**Keywords:** glial response, neurodegeneration, hippocampus, 5XFAD mice, immunohistochemistry, Alzheimer’s disease

## Abstract

In this study, the distinct patterns of glial response and neurodegeneration within the CA1, CA3, and dentate gyrus (DG) regions of the hippocampus were examined in 5XFAD mice at 6 and 12 months of age. The primary feature of this transgenic mouse model is the rapid onset of amyloid pathology. We employed quantitative assessments via immunohistochemistry, incorporating double staining techniques, followed by observation with light microscopy and subsequent digital analysis of microscopic images. We identified significantly increased Aβ deposition in these three hippocampal regions at 6 and 12 months of transgenic mice. Moreover, the CA1 and CA3 regions showed higher vulnerability, with signs of reactive astrogliosis such as increased astrocyte density and elevated GFAP expression. Additionally, we observed a significant rise in microglia density, along with elevated inflammatory markers (TNFα) in these hippocampal regions. These findings highlight a non-uniform glial and neuronal response to Aβ plaque deposition within the hippocampal regions of 5xFAD mice, potentially contributing to the neurodegenerative and memory deficit characteristics of Alzheimer’s disease in this model.

## 1. Introduction

For a prolonged span of years, neurons were regarded as the core functional elements of the central nervous system (CNS), whereas glial cells were thought to merely provide structural support. However, this perspective is undergoing a significant transformation; emerging evidence suggests that the intricate interplay among neurons, microglia, and astrocytes—the so-called “triad”—is crucial for the brain’s functional architecture and overall organization. Recent research has revealed that neuron–glial interactions are highly dynamic and can be significantly altered by inflammatory processes and other pathological insults [1,2,3,4]. Disruptions in neuron-astrocyte-microglia interactions may play a pivotal role not only in the normal brain aging process but also in the pathogenesis of neurodegenerative conditions, such as Alzheimer’s disease (AD) [5]. Therefore, a comprehensive understanding of these complex neuron–glia relationships is fundamental to the development of advanced therapeutic approaches for neurodegenerative disorders.

Astrocytes, previously considered solely supportive cells within the CNS, are now recognized for their intricate and dynamic roles, particularly in response to neuropathological conditions [6,7,8,9]. Although astrogliosis—a reactive change in astrocytes following CNS injury—has traditionally been associated with a loss of normal function and the promotion of neuronal damage, emerging research suggests that astrogliosis is a heterogeneous process, not always resulting in scar formation [10,11,12]. For example, a moderate level of reactive astrogliosis occurs in the CA3 hippocampal region, characterized by limited astrocyte proliferation that does not result in scar tissue but rather occupies non-overlapping domains [3,13]. The role of astrocytes in brain pathophysiology is, therefore, more intricate than previously understood. Astrocytes exhibit a range of cellular phenotypes, often initiating neuroprotective responses that are rarely purely harmful. Beneficial effects on neurons can result from adaptive reactive astrogliosis, while inhibiting astrocyte reactivity may enhance neuronal vulnerability, impair regeneration, and intensify disease progression [14,15,16]. Indeed, recent findings have challenged the earlier notion of microglial activation as a uniformly harmful process. It has become increasingly apparent that microglia are capable of responding to a diverse array of signals with a range of reactions, potentially exerting either neuroprotective or neurotoxic effects based on the specific stimuli they encounter. A growing body of research highlights the critical role of microglia in maintaining brain homeostasis, pruning synapses, phagocytosis of apoptotic neurons and cellular debris, and regulating astrocyte function [17,18,19,20,21,22,23]. Under acute or chronic neuroinflammation and neurodegeneration conditions, microglia are rapidly recruited to injury sites, where they adopt macrophage-like roles such as antigen presentation, phagocytosis, and release of immunomodulatory components [19,24,25]. However, microglial distribution and morphology vary significantly across different areas of the brain, such as the CA1 and CA3 regions of the hippocampus [26], resulting in region-specific susceptibilities to neurotoxicity or neuroprotection.

In AD mouse models, astrocytes are observed to increase around amyloid-beta plaques [27,28,29] and undergo atrophic changes characterized by reduced size and diminished branching [30,31,32]. However, astrocyte atrophy and astrogliosis do not appear uniformly across all brain regions, suggesting that astrocytes may exhibit different responses to similar stimuli in different areas, leading to varying effects on neuronal survival. Nevertheless, microglia typically act as the brain’s primary scavengers; their behavior changes in Alzheimer’s disease, where they proliferate and activate around amyloid-beta (Aβ) plaques. An impaired ability of microglia to properly clear Aβ is associated with an increased risk of developing AD [33]. If microglia are unable to respond to their regulatory feedback mechanisms or exhibit impaired Aβ clearance [34,35], they may transition to a predominantly cytotoxic state.

Moreover, microglia often show dystrophic, fragmented morphologies in aged brains, indicating that AD may arise as the neuroprotective functions of microglia diminish [36,37]. Additionally, there is considerable evidence indicating that excessive microglial activation can be detrimental to neurons, particularly in neurodegenerative diseases like AD [38]. Thus, strategically modulating microglial polarization at optimal times could offer therapeutic benefits for neurodegenerative conditions.

The hippocampus, a critical brain region for memory encoding, undergoes significant structural, morphological, and electrophysiological changes, particularly during aging and in neurodegenerative conditions such as AD [39,40,41,42,43,44]. These changes are especially prominent in the hippocampal subregions CA3, CA1, and DG, each of which has distinct anatomical characteristics and functions, playing a unique role in various types of information processing, including novelty detection, encoding, short-term and intermediate-term memory, and retrieval [45]. However, it is well established that the pyramidal neurons in the hippocampal CA1 region are particularly vulnerable to various insults, such as inflammation, hypoglycemia, ischemia, and excitotoxicity [46,47,48], in contrast to CA3 and dentate gyrus [49,50,51]. Some researchers attribute this increased sensitivity of CA1 neurons to the distinctive network properties inherent to the hippocampal formation [51], while others propose it may be due to the elevated density of NMDA receptors in this region [52]. Despite these theories, the exact cellular and molecular mechanisms behind this region-specific vulnerability remain unclear.

Nevertheless, it is essential to comprehend the neurobiological changes underlying hippocampal dysfunction and memory impairment to identify novel therapeutic targets aimed at preserving hippocampal function. There is still ambiguity regarding whether cellular responses to the same pathological conditions are uniform or vary across different hippocampal regions, such as CA1, CA3, and DG, and how these responses manifest. Furthermore, the specific roles of microglia and astrocytes in neurodegenerative diseases remain inadequately defined, with numerous and often contradictory aspects to consider. Therefore, this study aimed to assess and compare the quantitative alterations in the densities of astrocytes and microglia, emphasizing the distinct differences among the CA1, CA3, and DG regions of the brain hippocampus using 5xFAD transgenic mice as a model of AD relative to their wild-type (WT) counterparts. Advancing our knowledge of the specific roles that astrocytes and microglia play in the hippocampus’s CA1, CA3, and DG regions could offer valuable insights into the cellular and molecular mechanisms underpinning neurodegeneration in AD.

## 2. Results

In this experiment, images of H&E immunostaining in the CA1, CA3, and DG hippocampal regions were captured separately using light microscopy for quantitative analysis. Figure 1 shows the magnified views of the framed areas in the WT control, 6 M, and 12 M of 5XFAD mice.

### 2.1. Quantitative Analysis of Aβ Plaques Deposition in the Hippocampal CA1, CA3, and DG Regions of 5XFAD Mice

We performed immunohistochemistry to visualize the plaques in the hippocampal sections of 5xFAD mice at 6 and 12 months of age using anti-Aβ1-42 and anti-Aβ1-40 antibodies. The plaques ever found in the CA1, CA3, and DG regions of the hippocampus in WT control mice were negligible. Figure 2A and Figure 3A provide magnified views of the highlighted areas in the WT control, 6 M, and 12 M of 5XFAD mice.

The graph in Figure 2B represents the findings of the quantitative analyses of the Aβ1-42 staining density in the CA1, CA3, and DG hippocampal regions. We found a significant increase in Aβ1-42 density in all three brain regions at 6 and 12 months of 5xFAD mice in comparison to WT control. At 6 months, the most significant rise was seen in the DG region (+1859%) compared to WT control (** *p* < 0.001 vs. WT). On the other hand, the CA1 and CA3 regions exhibited a more moderate but still significant increase (+1418% and +828%, respectively) relative to WT mice (** *p* < 0.001 vs. WT).

At 12 months, the amyloid-β42 accumulation was even higher across all regions compared to both the WT and 6-month groups. The CA1 and CA3 regions displayed the highest significant increase in density (+4558% and +3016%, respectively) compared to WT mice (** *p* < 0.001 vs. WT). In the DG region, while the density of amyloid-β42 remained elevated (+2409%) compared to WT (* *p* < 0.05 vs. WT), it also showed a slightly lower increase compared to the CA1 and CA3 regions.

Moreover, the graph in Figure 3B represents the findings of the quantitative analyses of the Aβ1-40 density in the CA1, CA3 and DG hippocampal regions. A significant increase in Aβ1-42 density was observed in these hippocampal regions at 6 and 12 months of 5xFAD mice compared to WT control mice. At 6 months, the CA3 region showed the most significant rise (+857%) relative to WT control (** *p* < 0.001 vs. WT). In addition, a significant increase (+668%) was observed in the DG region (** *p* < 0.001 vs. WT), while the CA1 region showed a less significant increase (+353%) in the Aβ1-40 density (* *p* < 0.05 vs. WT) compared to WT control mice. By 12 months, the amyloid-β40 density further increased, with the DG region showing the most significant accumulation (+2300%) relative to WT, followed by the CA3 region by 1262% and the CA1 region by 820% (** *p* < 0.001 vs. WT).

### 2.2. Quantitative Analysis of Astrocytes in the Hippocampal CA1, CA3, and DG Regions of 5XFAD Mice

To perform a quantitative evaluation of astrocytes in the hippocampal sections of 5XFAD mice at 6 and 12 months of age and of WT control mice, immunohistochemistry was performed using an anti-GFAP antibody (as GFAP is a prototypical marker for astrocytes). Figure 4A illustrates the magnified views of the framed areas in the WT control, 6 M, and 12 M of 5XFAD mice.

The graph in Figure 4B represents the findings of the quantitative analyses of GFAP^+^ astrocytes in the CA1, CA3, and DG hippocampal regions. According to the result, the density of GFAP-positive astrocytes was significantly increased in all three brain regions of 5xFAD mice at 6 and 12 months of age compared to the WT control. Specifically, at 6 M, astrocyte density in the CA1 region increased significantly (+437%) compared to WT mice (** *p* < 0.001 vs. WT). However, the CA3 region showed an even more substantial increase (+996%) than WT (** *p* < 0.001 vs. WT). The astrocyte density also increased significantly (+423%) in the DG region (** *p* < 0.001 vs. WT).

Moreover, at 12 months of 5xFAD mice, GFAP-positive astrocyte density increased even further, particularly in the CA1 region where the density of astrocytes increased significantly (+898%) compared to WT control (** *p* < 0.001 vs. WT) and nearly five times from the 6 M level. The CA3 and DG regions also showed a modest increase from 6 M, increasing significantly (+1423% and +603%, respectively) compared to WT mice (* *p* < 0.05 vs. WT), which was still lower than the increases seen in CA1.

### 2.3. Quantitative Analysis of Microglia in the Hippocampal CA1, CA3, and DG Regions of 5XFAD Mice

For the quantitative assessment of microglia in the hippocampal sections of 5XFAD mice (6 M and 12 M) and control mice (WT), immunohistochemistry was performed using anti-IBA1 antibody (a specific marker for microglia). Figure 5A provides magnified views of the highlighted areas in the WT control, 6 M, and 12 M of 5XFAD mice.

The graph in Figure 5B represents the findings of the quantitative analyses of the density of IBA1^+^ microglia in the CA1, CA3, and DG hippocampal regions. The density of IBA1^+^ microglia was found to be increased in all three brain regions at 6 and 12 months of 5xFAD mice in comparison to WT control. At 6 M, there was a significant increase in microglial density in the CA1 region (+480%), CA3 region (+1506%), and DG region (+818%) compared to WT control (** *p* < 0.001 vs. WT).

By 12 M, the trend of elevated microglial density continued; in the CA3 region, the density increased significantly (+2010%) compared to WT (* *p* < 0.05 vs. WT), marking the most substantial rise among the regions. Meanwhile, the CA1 region also showed a significant increase (+734%) in the density of microglia compared to WT (** *p* < 0.001 vs. WT), whereas the density of IBA1^+^ microglia in the DG region increased (+731%), although not statistically significant compared to WT mice and showed a slight decrease from 6 M to 12 M.

### 2.4. Analysis of Inflammatory Mediators in the Hippocampal CA1, CA3, and DG Regions of 5XFAD Mice

To investigate the potential differences in the expression of inflammatory mediators among CA1, CA3, and DG regions, we performed H&E immunostaining using anti-TNF-α antibody on hippocampal sections of 5XFAD mice (6 M and 12 M) and control mice (WT). Figure 6A provides magnified views of the highlighted areas in the WT control, 6 M, and 12 M of 5XFAD mice.

The graph in Figure 6B represents the findings of the quantitative analyses of the density of TNF-α-positive cells in the CA1, CA3, and DG hippocampal regions. At 6 M, the CA1 region showed a significant increase (+64%) in TNF-α positive cell density (* *p* < 0.05 vs. WT), while the densities in the CA3 and DG regions remain almost similar to WT control. By 12 M, the increases in TNF-α positive cell density were more pronounced in hippocampal regions, with a significant increase in the CA1 region by 139% (** *p* < 0.001 vs. WT), CA3 region by 128% (* *p* < 0.05 vs. WT), and DG region by 221% (** *p* < 0.001 vs. WT) compared to WT control mice.

## 3. Discussion

The hippocampus is fundamental to memory formation and exhibits a wide range of electrophysiological, structural, and morphological modifications in response to both physiological and pathological states. Considerable advancements have been made in elucidating the connections between amyloid pathology and hippocampal dysfunctions, yet a comprehensive understanding of how these processes unfold across distinct hippocampal regions remains elusive.

Although often regarded as a distinct and cohesive structure, the hippocampus exhibits significant functional and molecular diversity across its various spatial domains. This unique synaptic and molecular architecture raises the intriguing question of how Alzheimer’s disease pathophysiology may differentially emerge in one hippocampal region compared to another. The main goal of this study was to investigate and compare the quantitative alterations in the interactions between astrocytes, microglia, and neurons within three primary hippocampal regions of interest (ROIs)—CA1, CA3, and DG in 5xFAD mice, a transgenic model of Aβ plaque formation, at 6 and 12 months old. This comparative approach holds significance due to the critical yet distinct roles these hippocampal regions play in memory processing and their prominent functional, structural, and morphological changes in AD [53]. Consequently, this study focused on elucidating the divergent patterns of neuronal degeneration, glial activation and alterations, and proinflammatory mediator expression at various phases of plaque accumulation. The key findings revealed that these adjacent and interconnected hippocampal regions exhibit strikingly distinct cellular responses and neuronal vulnerabilities in response to Aβ plaque accumulation.

It is well established that Aβ accumulation in Alzheimer’s disease causes neuronal dysfunction and cognitive deterioration, with memory impairment originating from disruptions in hippocampal synaptic function and progressively advancing toward extensive neurodegeneration and neuronal loss [54]. A growing body of literature also found that Aβ deposits in the CA1 and CA3 regions of the hippocampus indicate selective vulnerability in these areas, with the CA1 region being especially susceptible to neuronal loss and neurodegeneration [55,56,57], while CA3 typically being the least affected region [58,59]. Moreover, it has been evidenced that Aβ deposits are predominantly localized within the entorhinal cortex and the dentate gyrus of the hippocampus [60,61,62], supporting the notion of region-specific degeneration in AD. In line with this, our findings demonstrate a significant increase in Aβ deposits in the CA1, CA3, and DG hippocampal regions at 6 months of 5xFAD mice and continued to increase even more at 12 months of age. Conversely, it has been observed in various transgenic mouse models of AD that deficits in learning and memory do not consistently align with the accumulation of Aβ [63]. These findings indicate that memory dysfunction observed in AD is likely influenced by additional pathological mechanisms alongside Aβ deposition.

In our experiment, astrogliosis, characterized by an increased recruitment of astrocytes and elevated GFAP expression, was significantly more pronounced in the hippocampal CA1 region, particularly at 12 months in transgenic mice. However, this elevation in astrocyte density was considerably less prominent in the CA3 and DG regions compared to C1. In line with our findings, previous studies also noted age-related increases in astrocyte densities in 5xFAD and other transgenic mice models [64,65]. Astrocytes are essential not only for synaptogenesis, synaptic maturation, and maintenance [66,67] but also play a critical role in memory-related processes [68]. Nevertheless, they also participate in the progression of AD, a notion first introduced by Alois Alzheimer [69]. The role of astrocytes in AD progression varies according to the brain region affected and the severity of the disease. Within the hippocampus, AD has been linked to an early-stage astrocyte atrophy, known as clasmatodendrosis [70,71], which occurs in tandem with the appearance of reactive astrocytes surrounding amyloid plaques in later stages of the disease [31,32]. This duality underscores the complex role astrocytes play in both promoting synaptic health and contributing to neuroinflammation in the progression of AD.

Upon dissecting the hippocampus into three primary regions, CA1, CA3 and DG, significant differences were observed in various parameters investigated at distinct stages. In all three hippocampal regions of 6- and 12-month-old 5xFAD mice, we found TNF-α positive cells were notably elevated, aligning with previous findings in other neurodegenerative models [72,73]. TNF-α is primarily expressed in the brain by activated microglia and sometimes by astrocytes and neurons [74]. Furthermore, several investigations have demonstrated that the progressive accumulation of Aβ and the excessive production of inflammatory mediators stimulate astrocytes, leading to further increased synthesis and release of proinflammatory mediators, such as cytokines, interleukins, and nitric oxide (NO) [75,76]. These mediators subsequently initiate pro-apoptotic pathways in nearby brain regions [72], worsening Alzheimer’s pathology through enhanced neuroinflammation and cell death [77].

It is apparent that Aβ plaques trigger neurons and astrocytes to produce and release proinflammatory cytokines [78,79]. This cytokine release subsequently prompts a transition in microglial function from a surveillance or maintenance state toward executing immune responses. In general, microglia adopt two distinct phenotypes: the M1 phenotype, which is linked to the expression and release of proinflammatory cytokines [80,81], and the M2 phenotype, which plays a more protective role in maintaining tissue homeostasis, clearing apoptotic neurons, promoting regeneration and preventing secondary inflammation [82,83]. In our study of 5xFAD mice, we observed a notable increase in microglia density in all three hippocampal regions at varying degrees at 6 and 12 months of age. This suggests that Aβ plaques generate an inflammatory environment that induces enhanced microglial recruitment. A previous study with 5xFAD mice also supports our findings, where IBA1 immunostaining indicated similar age-dependent elevations in microglial densities, beginning at 8 months in the cortex and at 4 months in the hippocampus [64]. Moreover, the data reported by Ugolini et al. indicated that TgCRND8 mice at 3 and 6 months old showed similar age-dependent elevations in microglial presence in the hippocampal CA1 and CA3 regions [65].

Nevertheless, the reactive state of microglia, accompanied by increased astrogliosis, facilitates the formation of neuron-astrocyte-microglia triads. These cellular interactions allow astrocytes and microglia to collectively identify “danger signals”, such as cellular debris originating from apoptotic or necrotic neurons, and subsequently engage in the clearance of damaged neurons or neuronal fragments through phagocytosis [84]. Under normal physiological conditions, astrocytes and microglia exhibit protective roles, as they eliminate entire neurons via phagocytosis, clear dysfunctional synapses, and mitigate the spread of cellular damage while controlling inflammation [4]. Additionally, activated microglia help clear Aβ and cytotoxic debris from neural tissue. Conversely, it has been reported that microglia may also engage in the phagocytosis of living, healthy neurons in the inflamed central nervous system [85,86]. Prolonged microglial activity can exacerbate inflammation, promote further Aβ accumulation, and boost neurodegenerative processes [87]. Interestingly, microglia, similar to astrocytes, have been observed to exhibit dual roles in AD, either proinflammatory and harmful effects or anti-inflammatory and neuroprotective properties [88,89]. These diverse findings imply that depending on their location within the brain and the specific pathological conditions, microglia may exhibit distinct roles, functioning as either neuroprotective or neurotoxic agents.

Therefore, all these alterations in the brain hippocampal regions may serve as the underlying mechanisms for neurodegeneration and memory deficits, which are also observed in other transgenic mouse models of Aβ deposition, even in the early disease stages [90,91]. Interestingly, previous studies have demonstrated that enhanced physical activity mitigates cognitive and behavioral impairments in murine models of AD [92] and is related to a decreased risk of developing AD in humans [93]. Although the exact mechanisms behind the beneficial effects of physical activity remain unclear, astrocytes have been implicated as potential mediators [94]. This points to the possibility that targeting astrogliosis could emerge as a promising therapeutic strategy for AD [95], reflecting a growing interest in modulating neuroinflammatory and glial processes to ameliorate cognitive decline.

However, this study has several limitations and suggests improvements for future research directions to fully characterize glial responses in Alzheimer’s disease models. First, our analysis was confined to basic immunohistochemistry with hematoxylin/eosin counterstaining, which successfully met our research aim by allowing us to evaluate glial cell densities in targeted hippocampal regions. However, this approach lacked the specificity of additional markers that would allow for precise differentiation between the quiescent and activated states of astrocytes and microglia. Additionally, the absence of confocal fluorescence microscopy due to resource constraints limited high-resolution imaging, which is essential for detailed assessments of glial morphology and state across hippocampal regions. Future research should address these limitations by including a broader set of markers and advanced imaging techniques for a more refined understanding of glial responses. Furthermore, this study highlights the need to explore adult neurogenesis in the dentate gyrus (DG) within Alzheimer’s models in future studies. Investigating neurogenesis in the DG may shed light on how AD impacts neural stem cell populations and hippocampal function, potentially offering new insights into disease progression and therapeutic targets aimed at preserving cognitive functions through enhanced neurogenesis.

## 4. Materials and Methods

### 4.1. Animal Model

The 5xFAD transgenic mouse model, developed in 2006, carries five Familial Alzheimer’s disease (FAD) associated mutations: three in the human amyloid precursor protein (APP) gene (Swedish KM670/671NL, London V717I, and Florida I716V mutations) and two in the human presenilin 1 (PSEN1) gene (M146L and L286V mutations) [96]. These mutations significantly increase β-amyloid peptide production and aggregation, resulting in amyloid plaque formation and subsequent neurodegeneration [96]. Controlled by the Thy1 promoter, the transgenes are expressed primarily in neurons, as observed in human AD [97]. The hallmark of the 5xFAD model is its ability to replicate key features of AD, including β-amyloid plaque formation, synaptic dysfunction, neuroinflammation, and progressive cognitive decline [97]. Disease manifestations begin early, with intraneuronal β-amyloid deposits appearing by 1.5 months, followed by extracellular plaques at 2 months, particularly in brain regions like the subiculum, cortex, and hippocampus [64,98].

Two groups of transgenic mice, aged 6 and 12 months, were utilized in the study and compared to age-matched WT controls. Each transgenic age group consisted of five (n = 5) 5XFAD male mice, and their data were averaged for quantitative analysis. For the WT control, C57BL/6J mice were used at both 6 and 12 months (n = 10, equally divided by age). Since no significant differences were found between these two WT age groups across any of the parameters investigated in this experiment, their data were averaged to serve as a WT control group for comparative analysis. All animal experiments were performed in compliance with the Guiding Principles in the Care and Use of Animals (National Research Council, 1996) and approved by the Animal Experiment Ethics Committee of Keimyung University (KM-2022-01R1). All possible measures were taken to reduce animal suffering, and the number of animals used was kept to the absolute minimum required to obtain scientifically valid data. At the appropriate ages, mice were anesthetized using 5% isoflurane (JW Pharm, Seoul, South Korea) and perfused with 4% paraformaldehyde (in 0.1 M phosphate buffer, pH 7.4, 4 °C). Following overnight post-fixation and cryoprotection in 18% sucrose/PBS, 40 µm thick coronal brain sections containing the dorsal hippocampus were obtained using a cryostat and stored in antifreeze solution at −20 °C until immunohistochemistry procedures.

### 4.2. H&E and Immunohistochemistry Staining

The hippocampus was fixed using 10% paraformaldehyde in 0.1 M PBS (pH 7.4) and embedded in paraffin. Tissue sections were stained with hematoxylin-eosin (H&E) to evaluate overall tissue morphology and then subjected to immunohistochemical analysis. Paraffin blocks were sliced into 4–6 μm sections, which were then mounted on glass slides. The sections were deparaffinized using xylene and a series of ethanol dilutions, stained with H&E, and then immunostained to label cells that had migrated into the hippocampus. To quench endogenous peroxidase activity, all slides were incubated overnight at room temperature in 0.3% H_2_O_2_ in methanol. Primary antibodies, diluted at 1:200 to 1:1000 in PBS with 1% bovine serum albumin (BSA), were applied overnight at 4 °C. The slides were then incubated with HRP-conjugated secondary antibodies (diluted 1:500 in 5% BSA in PBS) at 37 °C for 1 h. Additionally, slides were stained with 1% Schiff’s reagent and Mayer’s hematoxylin for 5 min at room temperature. The proportion of immunostained positive cells in the tissue was compared to control results after three repeated experiments.

### 4.3. Antibodies

For immunohistochemistry, the following primary antibodies were used: For β amyloid plaque immunostaining: A rabbit anti-β amyloid 1-42 antibody, dilution 1:1000 (Product Code #ab201060, Abcam, Cambridge, UK); a rabbit anti-β amyloid 1-40 antibody, dilution 1:1000 (Product Code #44-136, Invitrogen, Waltham, MA, USA). For astrocytes: a rabbit anti-glial fibrillary acidic protein (GFAP) antibody, dilution 1:1000 (Product Code #ab68428, Abcam, Cambridge, UK). For microglia: a rabbit anti-ionized calcium binding adaptor molecule 1 (IBA1) antibody, dilution 1:1000 (Product Code #ab283319, Abcam, Cambridge, UK). For TNF-α: A rabbit anti-TNF-α antibody, dilution 1:500 (Product Code #ab1793, Abcam, Cambridge, UK). As secondary antibodies, mouse and rabbit IgG-HRP antibodies were used.

### 4.4. Microscopy Techniques and Quantitative Analysis

To perform quantitative analysis, images were captured at 40× magnification using an equipped slide scanner (MoticEasyScan One, Texas 78154, USA). Cells were meticulously counted, and the area of the analysis was measured. Cell density was then expressed as the number of cells per square millimeter (cells/mm^2^). All counting, measurements, and image analyses were conducted using ImageJ software (version 1.54g, National Institutes of Health, Bethesda, MD, USA). Using ImageJ, each image was first converted to an 8-bit grayscale format to simplify and standardize intensity measurement. Next, a specific threshold value was applied to identify and isolate the stained regions. Careful attention was given to maintaining consistent threshold settings for both control and transgenic slices from the same experimental batch. Once thresholded, the ImageJ software calculated the area of the positively stained regions, which was interpreted as the density of cells in the respective regions. The area exceeding the set threshold was calculated in pixels.

### 4.5. Statistical Analysis

The statistical analysis of the data was performed using a student’s *t*-test for independent methods using Microsoft Excel. A *p*-value of less than 0.05 was considered to be statistically significant. Data are presented as means ± SEM.

## 5. Conclusions

In conjunction with findings from other previous studies on the brain hippocampus under similar conditions, it can be concluded that the varying expression and distribution of astrocytes and microglia in the CA1, CA3, and DG regions may contribute to the differing vulnerability of these hippocampal areas to neurodegeneration during AD. Despite the anatomical proximity and interconnectivity of these regions in the mice hippocampus, our results highlight that glial responses to the same pathological stimuli are heterogeneous and vary considerably depending on the hippocampal region and under different stress conditions. Further investigation is warranted to target neuron–astrocyte–microglia triads, which may help to identify a potential therapeutic approach to modulate inflammatory responses and limit the propagation of cellular damage to adjacent cells, thereby mitigating the progression of neurodegenerative processes in AD.

## Figures and Tables

**Figure 1 ijms-25-12156-f001:**
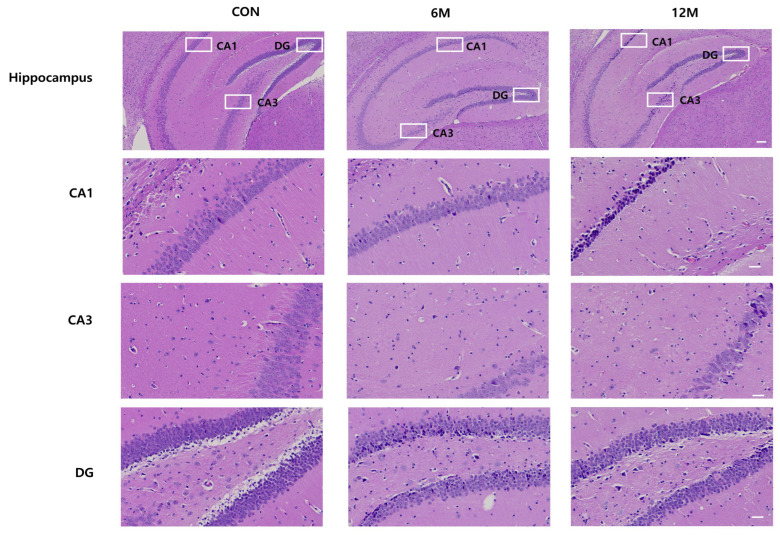
Representative sections of organotypic hippocampal slices of 5xFAD mice and WT control mice brains at 6 and 12 months of age stained with hematoxylin-eosin. (Scale bar 100 and 30 μm).

**Figure 2 ijms-25-12156-f002:**
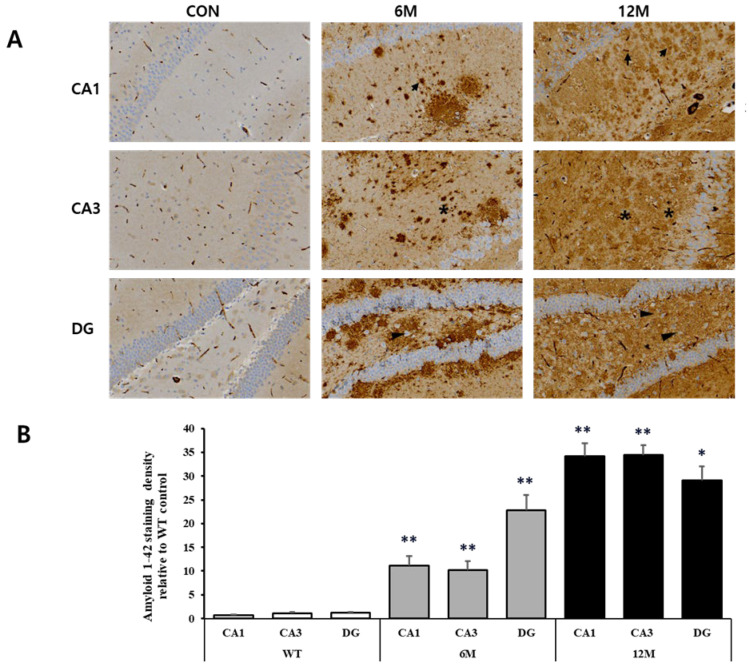
Analysis of Aβ1-42 plaques in the CA1, CA3, and DG regions of WT control, 6 M and 12 M 5XFAD Mice. (**A**) Representative sections of hippocampal organotypic slices stained with anti-Aβ1-42 antibodies and counterstained for hematoxylin/eosin. Arrows, asterisks, and arrowheads indicate positively immunostained cells. (**B**) Quantitative analysis of Aβ1-42-stained cells in the CA1, CA3, and DG hippocampal regions. The graph shows the density of Aβ1-42 staining in these hippocampal regions of 6 M and 12 M 5XFAD Mice relative to WT control. Data reported in all graph bars are expressed as mean ± SEM. ** *p* < 0.001 vs. WT control, * *p* < 0.05 vs. WT control. Scale bar: 30 μm.

**Figure 3 ijms-25-12156-f003:**
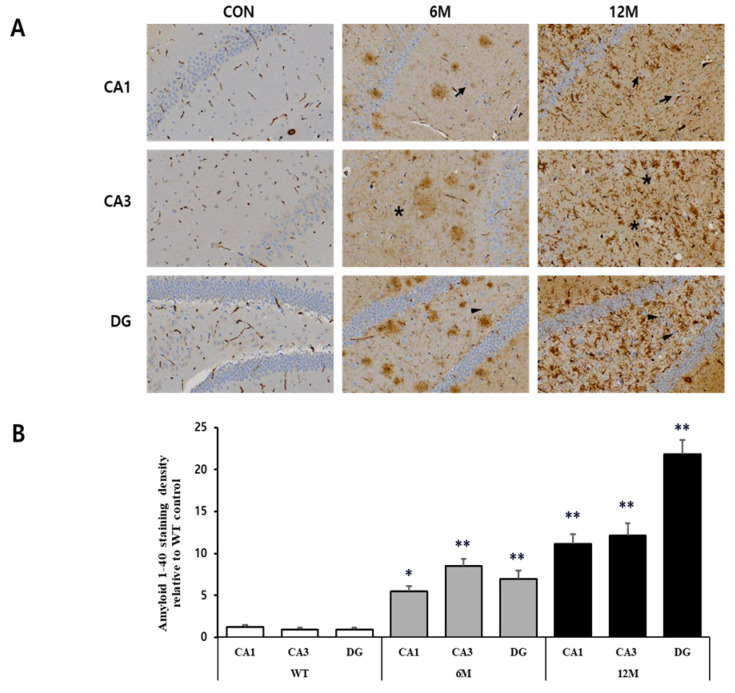
Analysis of Aβ1-40 plaques in the CA1, CA3, and DG regions of WT control, 6 M and 12 M 5XFAD Mice. (**A**) Representative sections of hippocampal organotypic slices stained with anti-Aβ1-40 antibodies and counterstained for hematoxylin/eosin. Arrows, asterisks, and arrowheads indicate positively immunostained cells. (**B**) Quantitative analysis of Aβ1-40-stained cells in the CA1, CA3, and DG hippocampal regions. The graph shows the density of Aβ1-40 staining in these hippocampal regions of 6 M and 12 M 5XFAD Mice relative to WT control. Data reported in all graph bars are expressed as mean ± SEM. ** *p* < 0.001 vs. WT control, * *p* < 0.05 vs. WT control. Scale bar: 30 μm.

**Figure 4 ijms-25-12156-f004:**
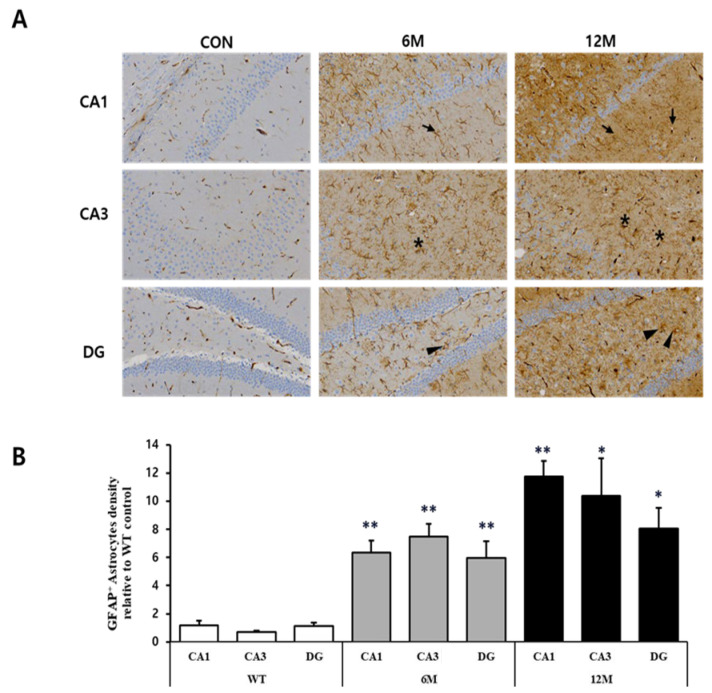
Analysis of astrocytes in the CA1, CA3, and DG regions of WT control, 6 M and 12 M 5XFAD Mice. (**A**) Representative sections of hippocampal organotypic slices stained with anti-GFAP antibodies and counterstained for hematoxylin/eosin. Arrows, asterisks, and arrowheads indicate positively immunostained cells. (**B**) Quantitative analysis of GFAP-positive astrocytes in the CA1, CA3, and DG hippocampal regions. The graph shows the density of GFAP-positive astrocytes in these hippocampal regions of 6 M and 12 M 5XFAD Mice relative to WT control. Data reported in all graph bars are expressed as mean ± SEM. ** *p* < 0.001 vs. WT control, * *p* < 0.05 vs. WT control. Scale bar: 30 μm.

**Figure 5 ijms-25-12156-f005:**
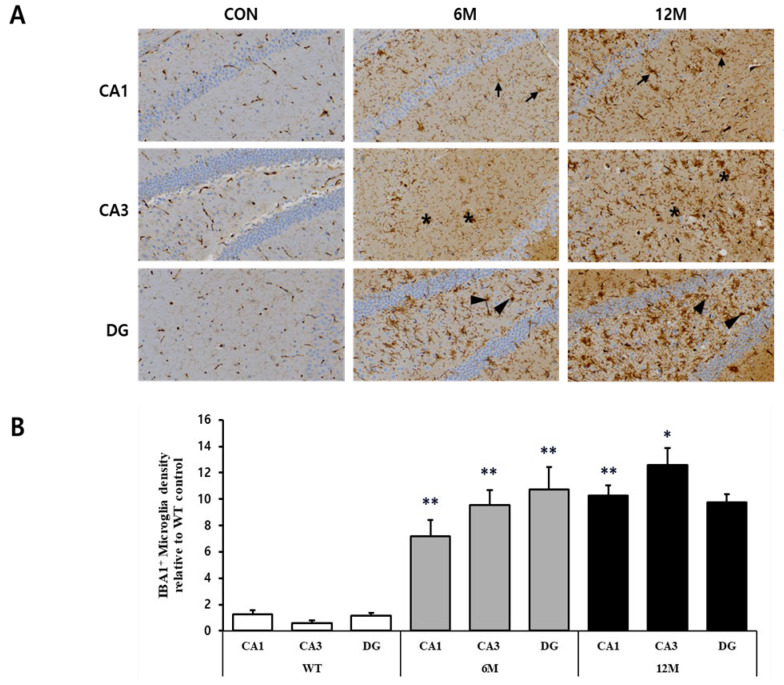
Analysis of microglia in the CA1, CA3, and DG regions of WT control, 6 M and 12 M 5XFAD Mice. (**A**) Representative sections of hippocampal organotypic slices stained with anti-IBA antibodies and counterstained for hematoxylin/eosin. Arrows, asterisks, and arrowheads indicate positively immunostained cells. (**B**) Quantitative analysis of IBA-positive microglia in the CA1, CA3, and DG hippocampal regions. The graph shows the density of IBA-positive microglia in these hippocampal regions of 6 M and 12 M 5XFAD Mice relative to WT control. Data reported in all graph bars are expressed as mean ± SEM. ** *p* < 0.001 vs. WT control, * *p* < 0.05 vs. WT control. Scale bar: 30 μm.

**Figure 6 ijms-25-12156-f006:**
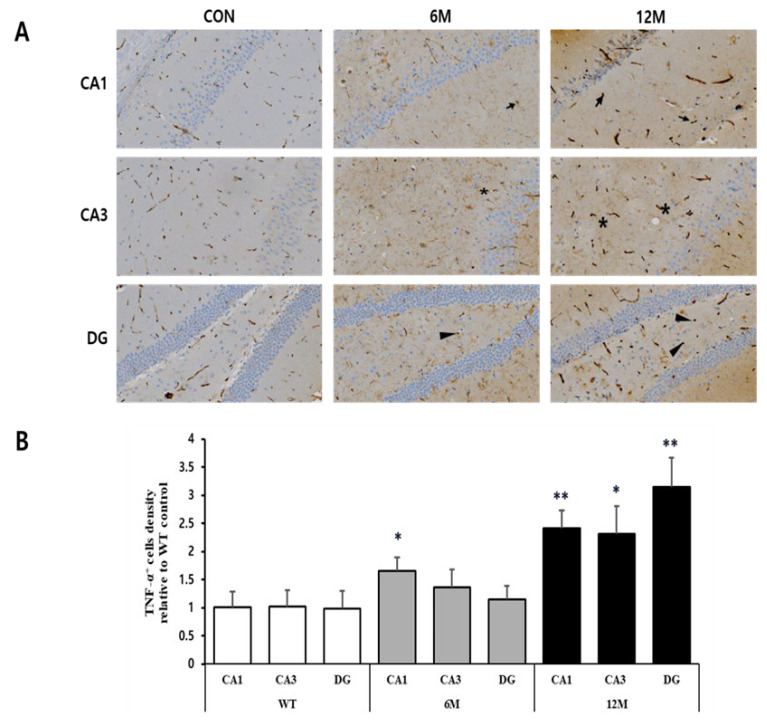
Analysis of TNF-α-positive cells in the CA1, CA3, and DG regions of WT control, 6 M and 12 M 5XFAD Mice. (**A**) Representative sections of hippocampal organotypic slices stained with anti-TNF-α antibodies and counterstained for hematoxylin/eosin. Arrows, asterisks, and arrowheads indicate positively immunostained cells. (**B**) Quantitative analysis of TNF-α-positive cells in the CA1, CA3, and DG hippocampal regions. The graph shows the density of TNF-α-positive cells in these hippocampal regions of 6 M and 12 M 5XFAD Mice relative to WT control. Data reported in all graph bars are expressed as mean ± SEM. ** *p* < 0.001 vs. WT control, * *p* < 0.05 vs. WT control. Scale bar: 30 μm.

## Data Availability

Data are contained within the article.

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
