# Peer review of "Differential Glial Response and Neurodegenerative Patterns in CA1, CA3, and DG Hippocampal Regions of 5XFAD Mice"

_ijms, 2024, doi:10.3390/ijms252212156_

Round 1

Reviewer 1 Report

Comments and Suggestions for Authors

The authors study the profiles of differentiated glial activation and neurodegeneration in 5xFAD AD mice model.

The study has serious defects which significant impair the science soundness of the study,

1, the authors use anti-GFAP antibody to determine activated astrocyte in mice brain. However, the GFAP can be the marker of activated and quiescent astrocytes. The antibody of anti-GFAP can only be used to check numbers of astrocyte. But can not tell activated and quiescent astrocyte. Furthermore staining with anti-GFAP antibody counterstained for hematoxylin/eosin is not a good practice to monitor astrocyte in brain slices. Confocal fluorescent study with higher resolutions should be used. The activated and quiescent astrocyte should be detected with different antibody. For quiescent astrocyte, AQP4, Aldh1L1 and GLAST should be detected. However, for activated astrocyte, Vimentin, Nestin, C3, Lcn2 and Serpina3n should be detected. 

2, similar comments on activated microglial cells results in Figure 5. The markers of quiescent microglial can be P2RY12, TMEM119, and CX3CR1, whereas activated microglial markers can be CD68, CD11b, and MHC Class II. The iba is the marker for both quiescent and activated microglial. Furthermore confocal fluorescent microscopy with high resolution and amplification times should be use, which can determine morphological changes to double confirm activated glial cells. 

3, the DG is the brain area to produce neuronal stem cells in adult. The authors should also go to check whether stem cell generation will be affects in the 5xFAD AD  mice model. 

Due to serious defects in their protocols and methods, the findings from the current study are seriously questioned, which can not be published at the current stage.

Author Response

Thank you for your valuable time and efforts. Please see the attachment.

Reviewer 2 Report

Comments and Suggestions for Authors

Nairuz et al. investigate glial activation, specifically GFAP-positive astrocytes and IBA1-positive microglia, in the CA1, CA3, and dentate gyrus regions of the hippocampus in 5XFAD mice.

However, this study lacks proper control samples, as age-matched WT control animals (6 and 12 months) should be included in Figures 1–6. The method used to quantify GFAP-positive astrocytes and IBA1-positive microglia appears to be arbitrary. How did you define the density of these cells?

Additionally, in Figure 6, there is no clear indication of TNF-α elevation in brain cells; the current representative images show TNF-α elevation predominantly in blood vessel-like structures, which does not support the conclusion of this study. 

Author Response

(The authors gave the same response as above.)

Reviewer 3 Report

Comments and Suggestions for Authors

The authors investigated differential glial activation and neurodegenerative patterns in CA1, CA3, and DG hippocampal regions of 5XFAD mice. The authors found a non-unform sensitivity to reactive astrogliosis with CA1 and CA3 the most impacted. The results has potential for high impact. The following are concerns on the manuscript in its present form:

1.      Sex is a major issue in 5xFAD mice as female mice show more aggressive response of the disease at much earlier age than their male counterparts. It is not clear which sex the authors used in their study.

2.      Do WT mice develop amyloid beta plaque pathology? What is the rational for quantifying a phenotype that does not exist.

3.      In figure 1 the hippocampus images it would have been better if the authors mark the regions that they magnified

4.      Did the authors use one control for 5XFAD at 6 and 12 months? There should have been two control comparing WT (6 &12) and 5XFAD (6 &12).

Author Response

(The authors gave the same response as above.)

Round 2

Reviewer 1 Report

Comments and Suggestions for Authors

The authors have addressed reviewer's comments properly. All concerns have been answered. I suggest accepting it in the current form.

Reviewer 2 Report

Comments and Suggestions for Authors

The authors have carefully addressed my comments. 

Reviewer 3 Report

Comments and Suggestions for Authors

I have no further concern on the manuscript.